# Enzymatic measurement of short-chain fatty acids and application in periodontal disease diagnosis

Kazu Hatanaka[1], Yasushi Shirahase[2], Toshiyuki Yoshida[2], Mari Kono[2], Naoki Toya[2], Shin-ichi Sakasegawa[3], Kenji Konishi[3], Tadashi Yamamoto[4], Kuniyasu Ochiai[4,5], Shogo Takashiba[4]*

1 Department of Periodontics and Endodontics, Okayama University Hospital, Okayama, Japan, 2 Sysmex Corporation, Kobe, Japan, 3 Asahi Kasei Pharma Corporation, Shizuoka, Japan, 4 Department of Pathophysiology-Periodontal Science, Dentistry and Pharmaceutical Sciences, Okayama University Graduate School of Medicine, Okayama, Japan, 5 Nihon University School of Dentistry, Chiyoda-ku, Tokyo, Japan

* stakashi@okayama-u.ac.jp

**Data Availability Statement:** All relevant data are within the article and its Supporting information files.

## Abstract

Periodontal disease is a chronic inflammatory condition caused by periodontal pathogens in the gingival sulcus. Short-chain fatty acids (SCFAs) produced by causal bacteria are closely related to the onset and progression of periodontal disease and have been reported to proliferate in the periodontal sulcus of patients experiencing this pathology. In such patients, propionic acid (C3), butyric acid (C4), isobutyric acid (IC4), valeric acid (C5), isovaleric acid (IC5), and caproic acid (C6), henceforth referred to as [C3–C6], has been reported to have a detrimental effect, while acetic acid (C2) exhibits no detrimental effect. In this study, we established an inexpensive and simple enzymatic assay that can fractionate and measure these acids. The possibility of applying this technique to determine the severity of periodontal disease by adapting it to specimens collected from humans has been explored. We established an enzyme system using acetate kinase and butyrate kinase capable of measuring SCFAs in two fractions, C2 and [C3–C6]. The gingival crevicular fluid (GCF) and saliva of 10 healthy participants and 10 participants with mild and severe periodontal disease were measured using the established enzymatic method and conventional gas chromatography-mass spectrometry (GC–MS). The quantification of C2 and [C3–C6] in human GCF and saliva was well correlated when using the GC–MS method. Furthermore, both C2 and [C3–C6] in the GCF increased with disease severity. However, while no significant difference was observed between healthy participants and periodontal patients when using saliva, [C3–C6] significantly differed between mild and severe periodontal disease. The enzymatic method was able to measure C2 and [C3–C6] separately as well as using the GC–MS method. Furthermore, the C2 and [C3–C6] fractions of GCF correlated with disease severity, suggesting that this method can be applied clinically. In contrast, the quantification of C2 and [C3–C6] in saliva did not differ significantly between healthy participants and patients with periodontal disease. Future studies should focus on inflammation rather than on tissue destruction.

**Funding:** This research was undertaken with a collaborative fund and nonfinancial support (sample analyses) from Sysmex Corporation (https://www.sysmex.co.jp/en/index.html) to Okayama University (S.T.; 2018 July – 2020 November). Four authors, Y.S., T.Y., M.K., N.T., collaborated this work as employees of Sysmex Corporation, and managed the contract research for this project, but did not have any additional role in the decision of publication. The specific roles of these authors are articulated in the 'author contributions' section. There was no additional external funding received for this study.

**Competing interests:** The authors have declared that no competing interests exist.

## Introduction

Periodontal disease is a chronic inflammatory condition caused by periodontal pathogens present in the gingival sulcus. Short-chain fatty acids (SCFAs) produced by causal bacteria have been reported to be closely related to the onset and progression of periodontal disease. SCFAs are monocarboxylic acids consisting of 2–6 carbons with one carboxyl group attached to an alkyl group and comprise of seven types: acetic acid (C2), propionic acid (C3), butyric acid (C4), isobutyric acid (IC4), valeric acid (C5), isovaleric acid (IC5), and caproic acid (C6), which do not include lactic acid and succinic acid [1]. These SCFAs are produced by bacteria in the process of metabolizing peptides (amino acids) and sugars into energy and have been widely studied as energy nutrients for mucosal epithelial cells in the colon, as well as activators and signal transmitters for immune cells in the intestinal mucosa [2]. However, SCFAs produced by the oral flora do not seem to function beneficially in the opposite direction. As in the intestinal tract, periodontal pathogenic bacteria produce a large quantity of SCFAs in the oral cavity [3]. However, Ochiai et al. reported that the production of C4 is several times higher than that of C2 in certain periodontal pathogenic bacteria, such as *Porphyromonas gingivalis* and *Fusobacterium nucleatum* and that C3, C4, and C5 induce the apoptosis of immune and nerve cells at high concentrations, while C2 does not have this extrinsic effect [4–6].

C4 has been reported to induce oxidative stress by spontaneous diffusion from periodontal tissues to the head and neck, and may also be involved in oxidative stress in the brain [7]. Furthermore, C3-C5 has been reported to reactivate various viruses, including human immunodeficiency virus (HIV), Epstein-Barr virus (EBV) and Kaposi's sarcoma-associated herpesvirus, suggesting that it may be deeply involved in the severity of periodontitis, as well as in systemic diseases [8–11]. The relationship between periodontal disease and systemic diseases, such as diabetes, obesity, coronary heart disease, stroke, rheumatoid arthritis, cancers, chronic kidney disease, preterm birth babies, respiratory disease, and Alzheimer's disease has been reported [12, 13]. The direct invasion of periodontal pathogenic bacteria into tissues and the elevation of cytokines due to inflammatory reactions due to proteases and lipopolysaccharides produced by periodontal pathogenic bacteria are thought to have a complex relationship [14].

We hypothesized that if we were able to establish a method to quantify SCFAs, particularly C3–C6 including C4 (henceforth referred to as [C3–C6]) and use it to determine the severity of periodontal disease, such a technique would contribute to the promotion of medical and dental coordination using the same index, as well as contribute to the treatment and prevention of a variety of systemic diseases. We previously established a method for the determination of trace amounts of SCFAs in saliva using conventional gas chromatography-mass spectrometry (GC–MS) that complies with the US Food and Drug Administration validation guidelines and reported the concentration distribution of SCFAs in saliva [15]. However, the clinical application of GC–MS has proven to be difficult owing to the complexity of its operation.

In this study, we aimed to develop an enzymatic reagent capable of measuring the distribution of SCFAs in the oral cavity using two enzymes (acetate kinase [AK] and butyrate kinase [BK]) in two fractions, C2 and [C3–C6], and measure SCFAs in the gingival crevicular fluid (GCF) and saliva of patients with periodontal disease using the enzymatic reagent. The correlation with the GC–MS method was examined to determine the clinical applicability of this assay technique using clinical specimens.

## Materials and methods

### Reagent

Adenosine diphosphate-dependent hexokinase (ADP-HK; from *Thermococcus litoralis*), glucose-6-phosphate dehydrogenase (G6PDH; from *Bacillus sp.*), and diaphorase (DI; from

*Bacillus megaterium*) were purchased from Asahi Kasei Pharma Corporation (Tokyo, Japan). Propionate kinase (PK; from recombinant *Escherichia coli*) was purchased from MyBioSource Incorporated (San Diego, CA, USA). Adenosine 5'-triphosphate disodium salt (ATP) and β-nicotinamide-adenine dinucleotide phosphate monosodium salt oxidized form (β-NADP) were purchased from Oriental Yeast Company, Limited (Tokyo, Japan). AK from *E. coli*, P1, P5-Di(adenosine-5') pentaphosphate pentasodium salt, valeric acid, isovaleric acid, and caproic acid were purchased from Sigma-Aldrich Corporation, Limited Liability Company (St. Louis, MO, USA). Magnesium chloride, glucose, and tris(hydroxymethyl)aminomethane isobutyric acid, octanoic acid and sodium succinate were purchased from Nacalai Tesque Incorporated (Kyoto, Japan). 2-(4-Iodophenyl)-3-(4-nitrophenyl)-5-(2,4-disulfophenyl)-2H-tetrazolium monosodium salt (WST-1) was purchased from Dojindo Laboratories (Kumamoto, Japan). Sodium acetate, sodium propionate, sodium butyrate, sodium lactate and sodium pyruvate were purchased from Fujifilm Wako Pure Chemical Corporation (Osaka, Japan). Sodium heptanoate was purchased from Tokyo Chemical Industry Company, Limited (Tokyo, Japan). The free acid of SFCAs was neutralized with sodium hydroxide and stored for later use. Normal commercial saliva was purchased from Lee Biosolutions, Incorporated (Maryland Heights, MO, USA).

## Overproduction and purification of recombinant His-tagged BK from *Thermosediminibacter oceani* (DSM 16646)

Chromosomal DNA from *T. oceani* was purchased from the Leibniz Institute, DSMZ-German Collection of Microorganisms and Cell Cultures GmbH (Braunschweig, Germany) and it served as the template. The entire BK gene was amplified using oligonucleotide primers 5'– ggcatatgac gttttacggg atactggcga –3' (sense) and 5'– gggaagcttt aaa tactcct ttggctttc –3' (antisense), which contain a unique *Nde*I and a unique *Hin*dIII restriction site, respectively (underlined). The 1,100-bp fragment of the BK gene was cloned into the corresponding sites in vector pET21a (+) to yield ToBKIII/pET21a (+), enabling His-tagged protein expression. *E. coli* BL21 (DE3) cells were transformed using ToBKIII/pET21a (+), and transformants were selected by growth on Luria-agar supplemented with ampicillin (50 μg/mL). The transformants were cultured in 1.6 L of Overnight Express™ Instant TB Medium (Merck, Darmstadt, Germany) containing 50 μg/mL ampicillin for 25 h at 30°C and pH 6.8 (650 rpm).

Cells expressing His-tagged BK were harvested by centrifugation, suspended in a 10 mmol/L potassium phosphate buffer (pH 7.5) and then disrupted by ultrasonication on ice. After removing the cell debris by centrifugation at 15,000 rpm for 20 min, His-tagged BK was purified using a Ni-chelating affinity column as described previously [16]. This purified protein (5 μg) was analyzed by sodium dodecyl sulfate-polyacrylamide gel electrophoresis (SDS-PAGE) using a 5%–25% gradient gel. After electrophoresis, the gel was stained with Coomassie Brilliant Blue R250.

BK activity was assayed as follows: the standard reaction mixture contained 50 mM Tris-HCl (pH 7.5), 2 mM ATP, 2 mM MgCl$_2$, 20 mM glucose, 100 mM sodium butyrate, 1 mM NADP, 5 U/mL glucose-6-phosphate dehydrogenase (Asahi Kasei, Tokyo, Japan), 5 U/mL ADP-dependent hexokinase (Asahi Kasei Corporation, Tokyo, Japan), and 0.1% TN-100 (Adeka Corporation, Tokyo, Japan) in a total volume of 150 μL. The reaction was initiated by the addition of 3 μL of the BK solution. The reaction mixture was incubated at 37°C in a cuvette, and the increase in absorbance at 340 nm was measured spectrophotometrically (d = 1 cm and ε340 = 6.3 mM$^{-1}$ cm$^{-1}$). In addition, 1 U of the enzyme was defined as the amount catalyzing 1 μmol of sodium butyrate per min at pH 7.5 at 37°C.

**Table 1. Background data of healthy controls and periodontitis patients.**

| Participant | Age | | Sex | | Specimen group | Sampling PD (mm) | Number of samples | |
|---|---|---|---|---|---|---|---|---|
| | Average | SD | M | F | | | GCF | Saliva |
| Healthy volunteers (healthy control) | 48.5 | 21.6 | 4 | 6 | HC | PD≤3 | 10 | 10 |
| Patients with mild periodontitis | 71.3 | 7.5 | 1 | 9 | PM | 3<PD≤5 | 10 | 10 |
| | | | | | PM Control | PD≤3 | 10 | - |
| Patients with severe periodontitis | 62.4 | 11.6 | 3 | 7 | PS | 5<PD | 10 | 10 |
| | | | | | PS Control | PD≤3 | 10 | - |

SD, standard deviation; M, male; F, female; GCF, gingival crevicular fluid; PD, probing depth; HC, healthy control; PM, mild group; PS, severe group

## Collection of specimens

Probing depth (PD) was measured in 1 mm increments. Ten patients with chronic periodontitis in the supportive periodontal therapy stage, who had two or more teeth with 3<PD≤5 mm (mild group [PM group]), 10 patients with two or more teeth with PD>5 mm (severe group [PS group]), and 10 healthy volunteers with PD≤3 mm (healthy control group [HC group]) were selected for the collection of saliva and GCF samples. The background characteristics are presented in Table 1. In addition, periodontal inflamed surface area (PISA) was also evaluated in some of the participants to validate periodontal inflammation from a clinical viewpoint because it is a very useful clinical index that expresses the surface area of the bleeding pocket epithelium in square millimeters and can be calculated by combining one of the conventional clinical parameters of periodontal disease, bleeding on probing, and PD [17].

Simultaneously, GCFs were collected from 2 healthy teeth (PD≤3 mm) of patients with periodontitis (PM control and PS control, n = 10 each). This study was conducted with the approval of the Ethics Committee, Okayama University Graduate School of Medicine, Dentistry and Pharmaceutical Sciences and Okayama University Hospital (Approved # 1807–025), and written informed consent was obtained from each participant after providing a written briefing of the contents of the study.

All saliva samples collected were secreted via stimulation after chewing gum for 5 min [18]. GCF was extracted by inserting three paper points (ZIPPERER® # 45, VDW GmbH, Munich, Germany) into a location on each tooth for 30 s and immersing 6 paper points for 2 teeth in 2 mL of saline [19]. A total of 50 GCF samples and 30 saliva samples were collected and stored at -80˚C. The samples were thawed and used on the measurement dates.

## Mechanism of reactivity employed by the enzymatic method

The following chemical reactions were used to convert the amount of SCFAs in saliva into the amount of β-NADPH for measurement (Formulae 1–3). The specific reaction could measure only a portion of SCFAs in the same reaction system (conjugation reaction of ADP-HK and G6PDH) by adding AK, PK, and BK separately (Table 2). Since the concentration of SCFAs was low in GCF, Formula 4 was added to increase the sensitivity.

$$
\begin{aligned}
&\textbf{AK}\\
&\text{Acetate (C2)} + \text{ATP} \quad \rightarrow \quad \text{Acetyl-phosphate} + \text{ADP}\\
&\textbf{PK}\\
&\text{Propionate (C3)} + \text{ATP} \quad \rightarrow \quad \text{Propyl-phosphate} + \text{ADP} \qquad (1)\\
&\textbf{BK}\\
&\text{Butyrate (C4)} + \text{ATP} \quad \rightarrow \quad \text{Butyryl-phosphate} + \text{ADP}
\end{aligned}
$$

**Table 2. Reagent composition and procedures for ultraviolet rays and color methods.**

| Handling | Test method | |
|---|---|---|
| | **Ultraviolet rays method (for saliva)** | **Color method (for GCF)** |
| R1 Reagent | 0.1 M Tris (pH 7.5) | 0.1 M Tris (pH 7.5) |
| | 20 mmol glucose | 40 mmol glucose |
| | 10 mmol MgCl2 | 20 mmol MgCl2 |
| | 2 mmol ATP | 4 mmol ATP |
| | 1 mmol β-NADP | 2 mmol β-NADP |
| | 0.1 mmol Ap5A | 0.2 mmol Ap5A |
| | 5 U/mL ADP-HK | 10 U/mL ADP-HK |
| | 5 U/mL G6PDH | 10 U/mL G6PDH |
| | | 20 U/mL AK or BK (or PK) |
| | | 0.8 mmol WST-1 |
| | | 10 U/mL DI |
| R2 Reagent | 0.1 M Tris (pH 7.5) | 1% Lauryl sulfate lithium |
| | 50 U/mL AK or PK or BK | |
| Standard | 2 mmol C2 or C3 or C4 in saline | 100 μmol C2 or C3 or C4 in saline |
| Procedure | After incubating 4 μL of the sample and 140 μL of R1 Reagent at 37˚C for 5 min, add 35 μL of R2 Reagent and measure the difference in absorbance before and after. | After adding 90 μL of the sample and 90 μL of R1 Reagent to the microplate, incubate at 25˚C for 15 min, add R2 Reagent and immediately measure the absorbance at 450 nm. |
| | Measure the C2, C3, or C4 standard solution in the same manner as the sample and compare it with the absorbance of the standard solution. | Similarly, blindly measure the absorbance of each sample using R2 Reagent excluding kinases and subtract from the first measured absorbance. |

ATP, adenosine 5'-triphosphate disodium salt; β-NADP, β-nicotinamide-adenine dinucleotide phosphate monosodium salt; ADP-HK, adenosine diphosphate-dependent hexokinase; G6PDH, glucose-6-phosphate dehydrogenase; AK, acetate kinase; BK, butyrate kinase; PK, Propionate kinase; WST-1, 2-(4-Iodophenyl)-3-(4-nitrophenyl)-5-(2,4-disulfophenyl)-2H-tetrazolium monosodium salt; C2, acetic acid; C3, propionic acid; C4, butyric acid; GCF, gingival crevicular fluid; Ap5A, P5-Di(adenosine-5') pentaphosphate pentasodium salt

$$\overset{\text{ADP} - \text{HK}}{\text{ADP} + \text{Glucose} \quad \rightarrow \quad \text{AMP} + \text{Glucose-6-phosphate}} \tag{2}$$

$$\overset{\text{G6PDH}}{\text{Glucose-6-phosphate} + \beta\text{-NADP}^+ \text{-} \quad \rightarrow \quad \text{6Phospho-Gluconate} + \beta\text{-NADPH}} \tag{3}$$

$$\overset{\text{DI}}{\text{B-NADPH} + \text{WST-1 (oxidized form)} \quad \rightarrow \quad \beta\text{-NAD} + \text{WST-1 (reduced Form)}} \tag{4}$$

## Measurement operation utilized by the enzymatic method

The reagents and methods used to measure SCFAs in saliva and GCF are shown in Table 2. In this composition, the ultraviolet rays (UV) method reacted at 37˚C for 10 min, and the color method reacted at 25˚C for 15 min. The saliva samples were measured by setting the R1 and R2 reagents of the UV method on a Hitachi 7180 autoanalyzer. Owing to the low concentration of SCFAs, the GCF was analyzed using the color method, which was not only

approximately four-fold more sensitive by adding diaphorase and WST-1, as shown in Formula 4, but also increased the minimum detection sensitivity by setting the sample to R1 reagent ratio to 1:1. The correlation of the two [C3–C6] enzyme measurement methods, UV and color, using commercial saliva is shown in Fig 1A. The saliva was 20-fold diluted with saline solution before use because of the high sensitivity of the color method.

The linearity of the [C3–C6] reagent using the color and UV methods for the butyric acid were performed as shown in Fig 1B and 1C. Regarding the substrate specificity of AK, PK, and BK, the organic acids were added to R2 reagent of the UV method at a concentration of 100 mM, and the enzyme activity was determined by adding AK, PK, and BK as samples at a concentration of 1 U/mL. The Michaelis–Menten constant (Km) was calculated by diluting 100

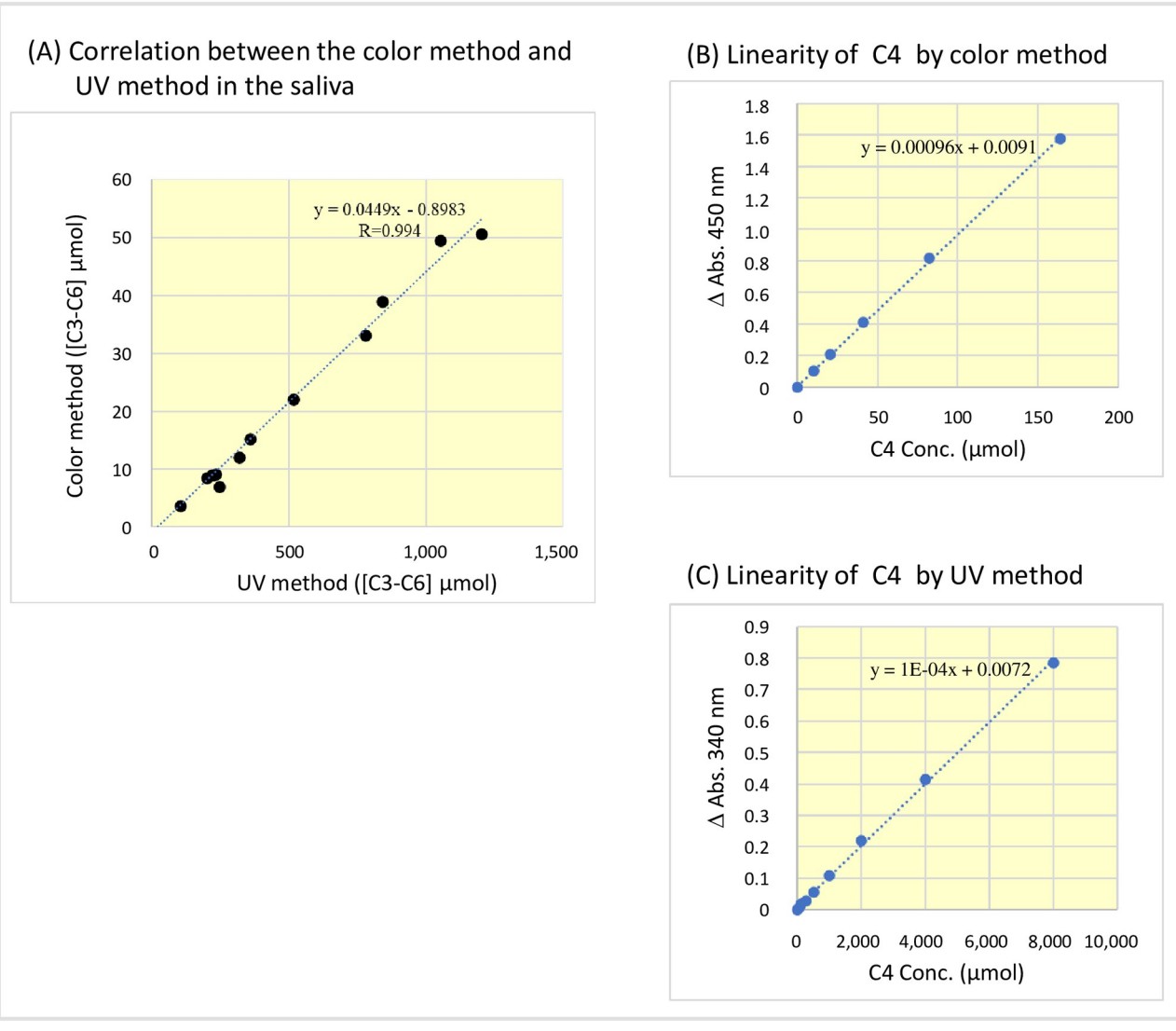

**Fig 1. Correlation between color method and ultraviolet rays method for [C3-C6] assay in the saliva and calibration curves for C4 obtained by these methods.** (A) Correlation between the vales obtained by color method and ultraviolet rays method for [C3-C6] standards. Since the color method is highly sensitive, saliva was diluted 20-fold with physiological saline for measuring. (B) Linearity of C4 calibration curve using the color method. (C) Linearity of C4 calibration curve using the ultraviolet rays method. [C3–C6], propionic acid, butyric acid, isobutyric acid, valeric acid, isovaleric acid, and caproic acid.

mM of the substrate in 0.1 mol Tris-HCl buffer (pH 7.5) to obtain the enzyme activity using R2 reagent [20].

## Measurement of SCFAs by GC–MS

The concentrations of SCFAs in the saliva and GCF of 30 participants were measured using the isotope dilution method using GC–MS.

Internal standards acetate-d3, propionate-d3 and n-butyrate-d7 solutions were prepared in a final concentration of 5 mmol/L. Lastly, iso-butyrate-1-13C, n-valerate-1-13C, and caproate-2,2-d2 solutions were prepared in a final concentration of 10 mmol/L.

Saliva and GCF samples (450 μL) were transferred to 1.5 mL micro-tubes. Ten μL of each internal standard solution was added, and the mixture was vortexed for 10 s. The newly mixed samples were deproteinized with 10 μL of a 5- sulfosalicylic acid solution [1 g/mL of ultra-pure water (w/v)] and centrifuged at 2,000 × g for 10 min at 4˚C. The supernatant was collected and transferred to a new test tube. Ten μL of hydrochloric acid (35%) and 3 mL of diethyl ether were added, by vortexing the mixture (30 min) after each addition. Then, the final mixture was centrifuged at 1,200 × g for 10 min at room temperature. The ether phase was retrieved and transferred to a new test tube. Next, tert-butyldimethylchlorosilane (8 μL) was added and the new solution heated at 60˚C for 20 min. After heating, the solution was cooled down and transferred to a glass vial for GC-MS analysis.

A gas chromatography-mass spectrometry apparatus (GCMS-QP2010 Ultra; Shimadzu, Kyoto, Japan), equipped with an autosampler (AOC-5000; Shimadzu, Kyoto, Japan), was used. The GC-MS equipment was run in chemical ionization mode, with ammonia as the reagent gas. Chromatographic separation was carried out using a DB-5 MS column (30 m × 0.25 mm I.D. × 0.25 μm; Agilent Technologies, Santa Clara, CA, USA). Helium (1.5 mL/min) was used as the carrier gas.

These measurements were performed at the Kyoto Institute of Nutrition & Pathology (Kyoto, Japan). More detailed operating procedures have been reported previously [15].

## Comprehensive analysis of bacterial flora by 16SrDNA present in the GCF

Bacterial DNA was extracted and purified from 50 GCF samples from 30 participants (HC, PM, PM control, PS, and PS control) using the QuickGene-810 system and QuickGene DNA tissue kit (KURABO) [21]. Primers, polymerase chain reaction (PCR) conditions, and purification procedures were prepared according to the 16S Metagenomic Sequencing Library Preparation provided by Illumina, and a sequence analysis was performed using Miseq (Illumina Incorporated, San Diego, CA, USA) based on the procedures provided by Illumina [22, 23]. The obtained text data were expressed as genus-level bacterial occupancy using quantitative insights into microbial ecology (QIIME: http://qiime.org).

## Statistical analysis

The difference in the mean C2 and [C3–C6] concentrations among the HC, PM, and PS groups were determined using the Mann–Whitney U test and are presented as box plots with quartiles. EZR software (version 1.41) was used for the statistical analyses.

## Results

### Biochemical characteristics of recombinant His-tagged BK from *T. oceani*

The properties and SDS-PAGE characteristics of the recombinant His-tagged BK are shown in S1 Table and S1 Fig., respectively.

## Correlation with the UV and color methods

We compared the correlation of [C3–C6] in the saliva between the UV method and the color method (20-fold dilution of the saliva in saline) and observed a correlation coefficient of 0.994 (Fig 1A).

## Reagent performance

The color method was confirmed up to 150 μmol (Fig 1B). Our results indicated that 1,000 μmol of C4 was >97% converted to β-NADPH after 3 min of reaction with R2 reagent and was completely converted to β-NADPH after 5 min, and linearity was confirmed up to 8,000 μmol (Fig 1C). The minimum detection sensitivity (LOD) was 65 μmol, and the coefficient of variation for reproducibility (CV) was 0.9% at 2,000 μmol.

## AK, PK, and BK substrate specificity

AK did not react with any reagent, in addition to C2. BK reacted with [C3–C6] with the same extent (Km, 2.1–8.7 mmol) regardless of carbon number, but not with C2. PK reacted incompletely reacted with C2, C4, and C3. No enzyme reacted with fatty acids or organic acids other than SCFAs (Table 3).

## Correlation with GC–MS

The correlation coefficients for C2 and [C3–C6] measured by the enzymatic method and GC–MS method for GCF were 0.850 and 0.922, respectively, which were lower than those for saliva (Fig 2A and 2B). The correlation coefficients for C2 and [C3–C6] measured by both methods for saliva were better than 0.986 and were on the straight line of the regression equation from 100–6,000 μmol, with deviations in the samples (Fig 2C and 2D).

## Application in periodontal disease examination

The results of the Mann–Whitney U test for the mean values among the HC, PM, and PS groups by specimen type are shown in Fig 3. C2 and [C3–C6] in the GCF increased in

**Table 3. Substrate specificity of AK, PK, and BK.**

| organic acids | AK | | PK | | BK | |
|---|---|---|---|---|---|---|
| | Relative activity (%) | Km (mmol) | Relative activity (%) | Km (mmol) | Relative activity (%) | Km (mmol) |
| C2 | 100 | 9.1 | 46 | 13.2 | 2 | - |
| C3 | 2 | - | 100 | 1.8 | 90 | 5.1 |
| C4 | 0 | - | 9 | 280 | 100 | 2.2 |
| IC4 | 0 | - | 0 | - | 96 | 3.2 |
| C5 | 0 | - | 0 | - | 99 | 3.3 |
| IC5 | 0 | - | 0 | - | 96 | 2.1 |
| C6 | 0 | - | 0 | - | 84 | 8.7 |
| C7 | 0 | - | 0 | - | 5 | - |
| C8 | 0 | - | 0 | - | 1 | - |
| Lactate | 0 | - | 1 | - | 1 | - |
| Pyruvate | 1 | - | 0 | - | 0 | - |
| Succinate | 0 | - | 0 | - | 0 | - |

AK, acetate kinase; BK, butyrate kinase; PK, Propionate kinase; C2, acetic acid; C3, propionic acid; C4, butyric acid; IC4, isobutyric acid; C5, valeric acid; IC5, isovaleric acid; C6, caproic acid (C6); C7; heptanoic acid (C7); C8, caprylic acid (C8)

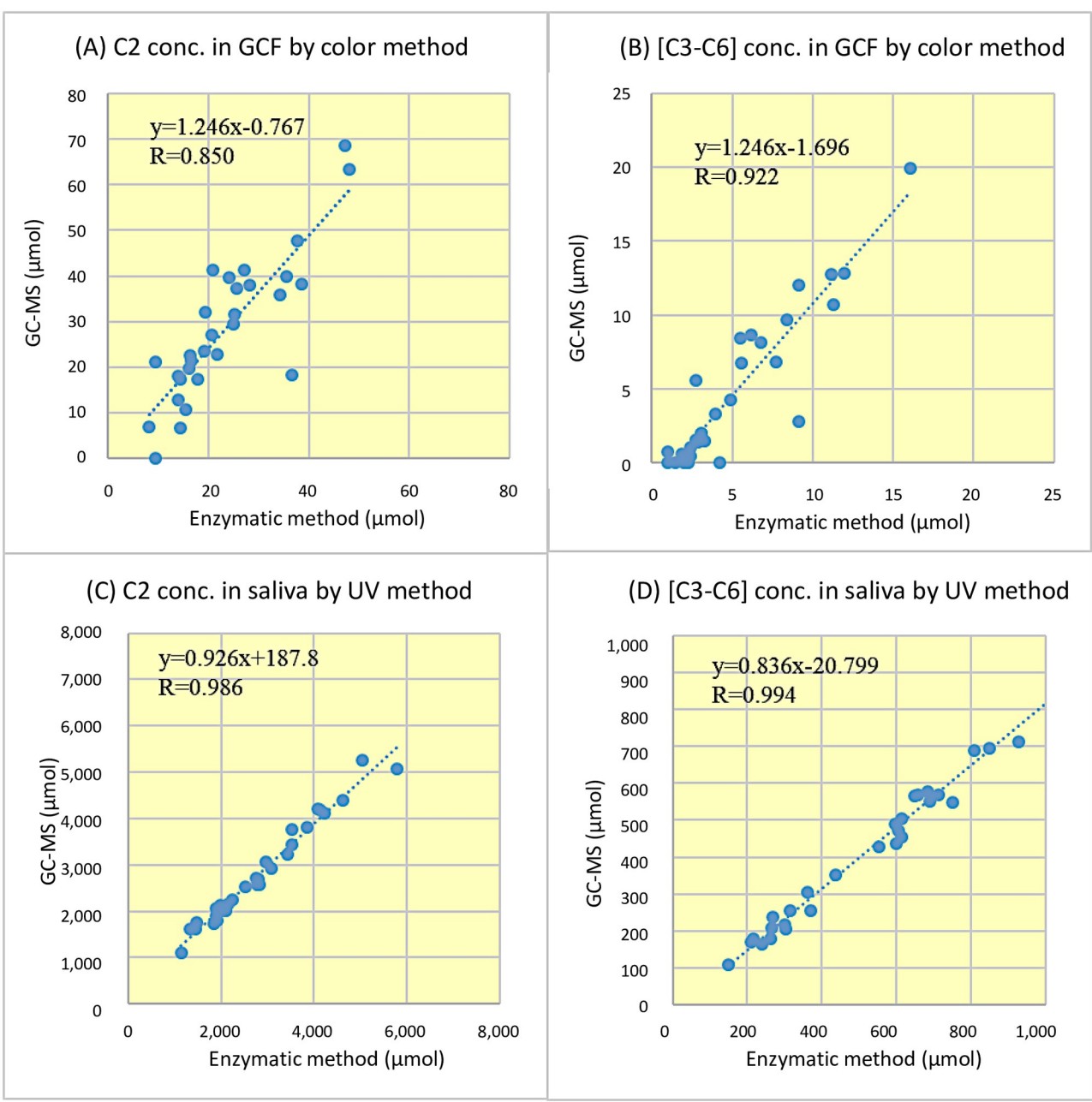

**Fig 2. Correlation between the enzymatic method and GC–MS method for the measurement of C2 and [C3–C6] in the GCF and saliva.** (A) C2 and (B) [C3-C6] in the GCF were measured using the color method of the enzymatic method. (C) C2 and (D) [C3-C6] in the saliva were measured using the ultraviolet rays method of the enzymatic method. C2, acetic acid; [C3–C6], propionic acid, butyric acid, isobutyric acid, valeric acid, isovaleric acid, and caproic acid; GCF, gingival crevicular fluid; GC–MS, conventional gas chromatography-mass spectrometry.

proportion to the severity of periodontal disease (A: HC vs. PM, $p < 0.05$; PM vs. PS, $p < 0.01$; HC vs. PS, $p < 0.001$ for C2, and B: PM vs. PS, $p < 0.01$; HC vs. PS, $p < 0.001$ for [C3–C6]). Although no significant difference in the saliva was observed among the HC, PM, and PS groups in proportion to the severity of periodontal disease, there was a significant difference in [C3–C6] between the PM and PS groups ($p < 0.05$).

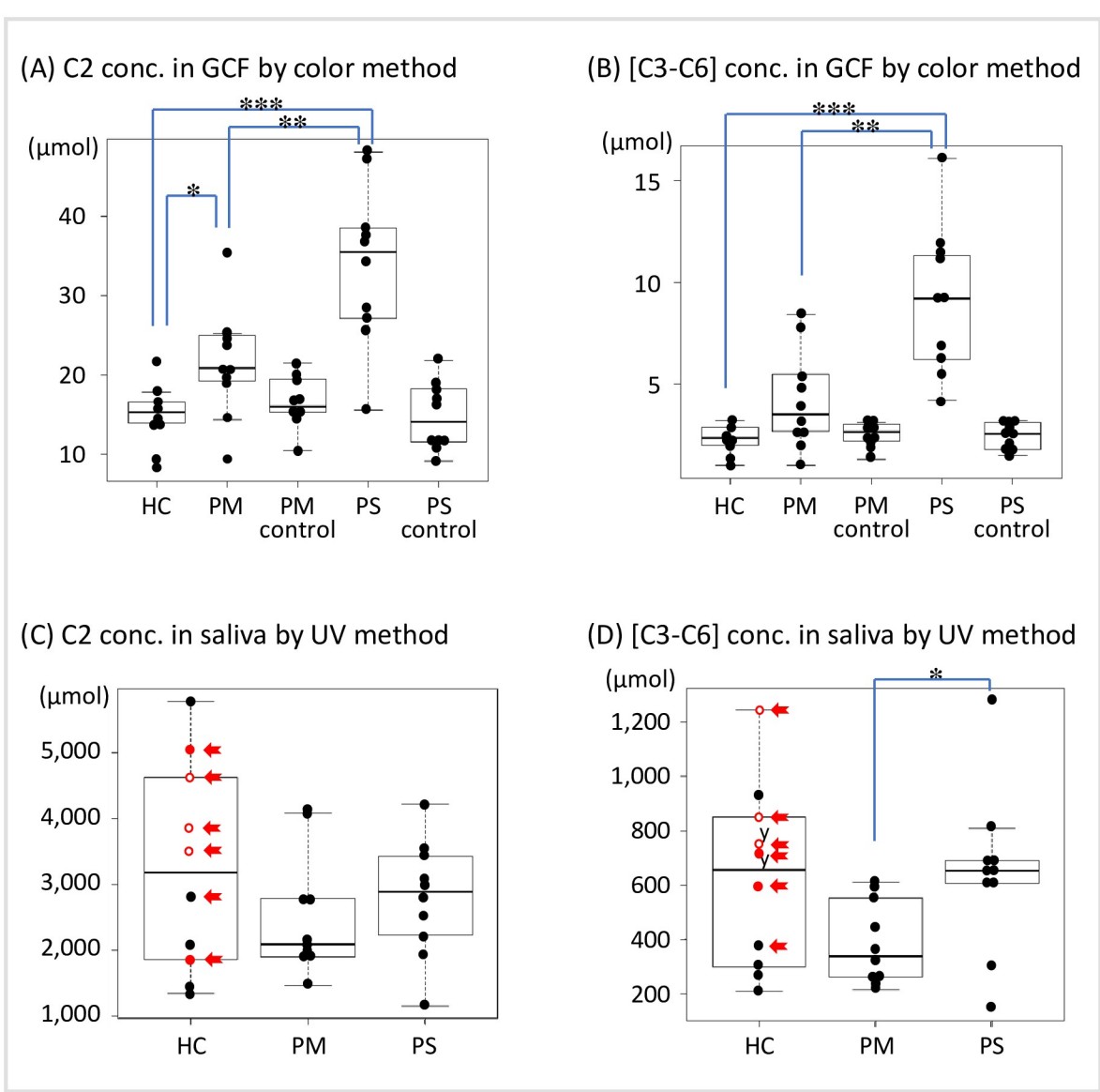

**Fig 3. Concentration distribution of C2 and [C3–C6] in periodontal disease severity.** (A) C2 and (B) [C3–C6] in the GCF were measured using the color method. (C) C2 and (D) [C3–C6] in the saliva were measured using the ultraviolet rays method as in Fig 2. C2 and [C3–C6] concentrations in the saliva and GCF are shown in a boxplot with the interquartile range. Mann–Whitney U test; *: $p < 0.05$, **: $p < 0.01$, *** $p < 0.001$. The samples of PM control and PS control shown in (A) and (B) are GCF samples collected from healthy teeth (PD≤3 mm) from the PM and PS groups. The red closed dots indicate a PISA>100 mm², and the open dots indicate a PISA>300 mm² after 1.5 years. The red arrow indicates the sample in which *Porphyromonas gingivalis* was present in the GCF. C2, acetic acid; [C3–C6], propionic acid, butyric acid, isobutyric acid, valeric acid, isovaleric acid, and caproic acid; GCF, gingival crevicular fluid; PM, mild group; PS, severe group; PD, probing depth; PISA, periodontal inflamed surface area.

## Investigation of the cause of the high [C3–C6] deviation in the HC group

The results of the bacterial flora analysis in the GCF samples from healthy participants are described in S2 Fig. In the specimens of healthy participants with PD ≤ 3 mm, those in whom *P. gingivalis* was detected (6 GCF samples in the red box in S2A Fig) exhibited high concentrations of C3-C6 (5 out of the 6 saliva samples, S2B Fig).

## Discussion

We developed a reaction system that could measure SCFAs in two fractions, C2 and [C3–C6], using AK and BK (Table 2), and confirmed that the correlation between this method and GC–MS was equivalent using the GCF and saliva from clinical samples (Fig 2). Next, we applied the method to determine the differences in the pathogenesis of the periodontal disease, and the results for the GCF reflected the differences in the pathogenesis of C2 and [C3–C6] (Fig 3A and 3B). On the contrary, for the saliva, we could not classify the healthy group and the periodontal disease group, but we could distinguish between the PM and PS groups for [C3–C6] (Fig 3C and 3D).

In general, the GC–MS method used to measure SCFAs is complicated and time-consuming [24], making it difficult to use in clinical practice. In contrast, the newly developed enzyme method can efficiently measure numbers of samples in a short procedure as shown in Table 2. It is also environmentally friendly because it does not use highly toxic organic solvents. The ability to distinguish between C2 and [C3-C6] simply by using the enzyme (Table 3) and the good correlation with the conventional standard GC-MS method using clinical samples such as GCF and saliva (Fig 2) indicates that this is a simple, inexpensive, and quick method that can be applied in clinical settings (Fig 3).

In the GCF, which reflects local inflammatory conditions, C2, [C3–C6], in that order, reflected the differences in pathology between the HC and PM groups and the PS group. In the saliva, [C3–C6] reflected the differences in pathology between the PS and PM groups better than C2, as reported by Ochiai et al. [4]. However, for the saliva, there was no significant difference between the HC and PM groups and the PS group. To investigate the cause of this saliva deviation, we checked the PISA values, which reflect clinical inflammatory conditions based on periodontal pocket probing of the six samples that deviated to high values in the HC group, and found that five samples were above 100 mm$^2$ (Fig 3C and 3D). Furthermore, the same five samples contained *P. gingivalis*, which produces a large number of SCFAs in the GCF flora (S2 Fig). Previously, the bacterial flora in the saliva of 977 Japanese individuals was analyzed, and a strong correlation was found between the presence of *P. gingivalis* and periodontitis [25]. These findings suggest that the HC group included a group of preliminary periodontitis samples that already exhibited inflammation even though periodontal tissue destruction expressed by PD was normal, and that the high prevalence of *P. gingivalis*-related inflammation in these samples may be the reason for the lack of difference between the HC group and the PM or PS groups.

The measurement of SCFA may be a better indicator of the degree of inflammation than the degree of periodontal tissue destruction. If the degree of inflammation in saliva can be measured using this method, it is likely that the relationship between periodontal disease and inflammation-affected conditions, such as diabetes, can be detected more sensitively than with conventional periodontal examinations that measure the degree of destruction. There is also the possibility of using this method to make devices, such as test strips for urinalysis, which could be used to determine the severity of the disease on the spot. In the future, if we can determine the severity of periodontitis using the same index in both medicine and dentistry by examining the sample collection method, we believe that we can greatly contribute to the mode of cooperation between the two fields. Although enzymatic measurement such as color method for GCF (low concentration) and UV method for saliva (high concentration) seemed to be superior than the GC-MS measurement, further large-scale study with balance between sex and age groups should be performed to concrete our proof-of-concept study. Furthermore, because the severity of periodontitis was determined by periodontal pocket depth, the results were favorable for the GCF, which reflects local conditions, but not for the saliva, which is

affected by the entire oral cavity. In this point, it will be necessary to consider indices, such as high-sensitivity C-reactive protein levels, which indicates the degree of inflammation in periodontitis [26, 27].

## Conclusion

The enzymatic method developed in this study is a simple, inexpensive, and rapid method, and the measurement of SCFAs in the GCF and saliva by this method can be clinically applied to screen the severity of periodontal disease.

## Supporting information

**S1 Table. Biochemical properties of recombinant His-tagged butyrate kinase from *Thermosediminibacter oceani* (DSM 16646).**
(DOCX)

**S1 Fig. Sodium dodecyl sulfate-polyacrylamide gel electrophoresis analysis of butyrate kinase from *Thermosediminibacter oceani*.**
(TIF)

**S2 Fig. Additional information for the healthy control group.**
(TIF)

**S1 Raw image.**
(TIF)

**S1 Data.**
(XLSX)

## Acknowledgments

We would like to thank our dental staff for supporting for obtaining clinical samples and for having useful discussions.

## Author Contributions

**Conceptualization:** Yasushi Shirahase, Shogo Takashiba.

**Data curation:** Kazu Hatanaka, Yasushi Shirahase.

**Formal analysis:** Yasushi Shirahase, Naoki Toya.

**Funding acquisition:** Toshiyuki Yoshida, Mari Kono.

**Investigation:** Kazu Hatanaka, Yasushi Shirahase, Shogo Takashiba.

**Methodology:** Kazu Hatanaka, Yasushi Shirahase, Mari Kono, Shogo Takashiba.

**Project administration:** Toshiyuki Yoshida.

**Resources:** Shin-ichi Sakasegawa, Kenji Konishi.

**Software:** Naoki Toya.

**Supervision:** Mari Kono.

**Validation:** Yasushi Shirahase, Mari Kono, Tadashi Yamamoto, Kuniyasu Ochiai.

**Visualization:** Yasushi Shirahase, Naoki Toya.

**Writing – original draft:** Kazu Hatanaka, Yasushi Shirahase, Shogo Takashiba.

**Writing – review & editing:** Kazu Hatanaka, Yasushi Shirahase, Mari Kono, Tadashi Yamamoto, Kuniyasu Ochiai, Shogo Takashiba.

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
