## [Decision Letter · Decision Letter 0]

28 Mar 2022

PONE-D-21-34023Enzymatic measurement of short-chain fatty acids and application in periodontal disease diagnosisPLOS ONE

Dear Dr. Takashiba,

Thank you for submitting your manuscript to PLOS ONE. After careful consideration, we feel that it has merit but does not fully meet PLOS ONE’s publication criteria as it currently stands. Therefore, we invite you to submit a revised version of the manuscript that addresses the points raised during the review process.

We look forward to receiving your revised manuscript.

Kind regards,

Joseph Banoub, Ph,D., D. Sc.

Academic Editor

PLOS ONE

Journal Requirements:

This research was undertaken with collaborative funds and nonfinancial support (sample analyses) from Sysmex Corporation (https://www.sysmex.co.jp/en/index.html). Four authors, Y.S., T.Y., M.K., N.T., performed this work as employees of Sysmex Corporation, and managed the contract research for this project.

Author Contributions are shown below.

Conceptualization: Y.S., S.T.

Data curation: K.H., Y.S.

Formal analysis: Y.S., N.T.

Funding acquisition: T.Y., M.K.

Investigation: K.H., Y.S., S.T.

Methodology: K.H., Y.S., M.K., S.T.

Project administration: T.Y.

Resources: S.S., K.K.

Supervision: M.K.

Validation: Y.S. M.K., T.Y., K.O.

Visualization: Y.S., N.T.

Writing – original draft: K.H., Y.S., S.T.

Writing – review & editing: K.H., Y.S., M.K., T.Y., K.O., S.T.

I have read the journal's policy and the authors of this manuscript have the following competing interests: Sysmex Corporation for collaborative funds and nonfinancial support.  Four authors, Y.S., T.Y., M.K., N.T., performed this work as employees of Sysmex Corporation.

7. We note that you have included the phrase “data not shown” in your manuscript. Unfortunately, this does not meet our data sharing requirements. PLOS does not permit references to inaccessible data. We require that authors provide all relevant data within the paper, Supporting Information files, or in an acceptable, public repository. Please add a citation to support this phrase or upload the data that corresponds with these findings to a stable repository (such as Figshare or Dryad) and provide and URLs, DOIs, or accession numbers that may be used to access these data. Or, if the data are not a core part of the research being presented in your study, we ask that you remove the phrase that refers to these data.

Reviewers' comments:

Reviewer's Responses to Questions

**Comments to the Author**

1. Is the manuscript technically sound, and do the data support the conclusions?

Reviewer #1: Yes

Reviewer #2: Partly

2. Has the statistical analysis been performed appropriately and rigorously? 

Reviewer #1: Yes

Reviewer #2: Yes

3. Have the authors made all data underlying the findings in their manuscript fully available?

Reviewer #1: Yes

Reviewer #2: Yes

4. Is the manuscript presented in an intelligible fashion and written in standard English?

Reviewer #1: Yes

Reviewer #2: Yes

5. Review Comments to the Author

Reviewer #1: In the manuscript titled “Enzymatic measurement of short-chain fatty acids and application in periodontal disease diagnosis”, Hatanaka et al. developed and examined an enzymatic assay for the assessment of short-chain fatty acids that may be used for periodontal disease. Specific comments are as follows:

1) For the examination of patient samples, it needs to be stressed that the number of subjects was small and thus the use of the enzymatic assay is exploratory at this point. More subjects would be needed for all groups, with balance between sexes and age groups. A discussion of this limitation is needed.

2) While the UV and colorimetric assessments of the enzymatic assay exhibit linearity, it is unusual that colorimetric data were only provided for Fig. 1A and 1B, and UV data for Fig. 1C and 1D. For completeness, also provide UV data (to accompany assessed samples for 1A and B) and colorimetry data (to accompany assessed samples for 1C and 1D).

Reviewer #2: Really interesting work.

But :

- Please do not use supporting information to store information that can bring important data to understand the problems and discovery, especially in results part, concerning S2B fig and the correlation with UV and color method (line 287)

- describe GC-MS method, by giving the conditions and material used for these analyses, instead of giving a reference

- include lines 277-282 to the figure 1 legend. Same thing for line 299-310

- line 329-333 : this conclusion seems to be unsuitable, since it depends the precision we are looking for. in fact, if in saliva the comparaison betweem GC-MS and enzymatic methods lead to an almost good correlation, it is note the case in GCF.

- in the conclusion, it should be interesting to give more information in term of saving money and time

6. PLOS authors have the option to publish the peer review history of their article (what does this mean?). If published, this will include your full peer review and any attached files.

Reviewer #1: No

Reviewer #2: No

---

## [Author Response · Author response to Decision Letter 0]

27 Apr 2022

To Editor:

We clarified the funding resources and funder's roles by correcting the "Funding acquisition", because the corresponding author (S.T.) at Okayama University received the collaborative funds from Sysmex Corporation, for which Yasushi Shirahase (Y.S.), Toshiyuki Yoshida (T.Y.), Mari Kono (M.K.), and Naoki Toya (N.T.) work as employees. They collaborated to this work with their roles as listed in the "Author Contributions" section. Please refer our updated description at the bottom of this letter.

In addition, we would like to address to additional requirements.

1. Manuscript style and file names. We corrected them according to journal guideline.

2. ‘Funding Information’ and ‘Financial Disclosure’. We corrected as descried above.

3. Financial disclosure. Again, we corrected as descried above. We amend it at the bottom of this letter.

4. Competing Interests. Although this study was supported by Sysmex Corporation for a collaborative fund and nonfinancial support, this does not alter our adherence to PLOS ONE policies on sharing data and materials. Updated Competing Interests statement can be found at the bottom of this letter.

5. Minimal data set. The minimal data set is now available as a Supporting Information file.

6. Original uncropped and unadjusted images. The original gel image is now available with appropriate legend as a Supporting Information file.

7. The phrase “data not shown”. This issue has been dissolved by showing this data as Fig 1B.

To Reviewer #1

In the manuscript titled “Enzymatic measurement of short-chain fatty acids and application in periodontal disease diagnosis”, Hatanaka et al. developed and examined an enzymatic assay for the assessment of short-chain fatty acids that may be used for periodontal disease. Specific comments are as follows:

<Response>

Thank you for your valuable comments. We carefully reconsider our manuscript and sincerely respond to your comments.

1) For the examination of patient samples, it needs to be stressed that the number of subjects was small and thus the use of the enzymatic assay is exploratory at this point. More subjects would be needed for all groups, with balance between sexes and age groups. A discussion of this limitation is needed.

<Response>

We completely agree with your concern that this study has the limitation for subjects, i.e., number, sex, age, and so on. We added the discussion on these limitations in Discussion section as follows.

Although enzymatic measurement such as color method for GCF (low concentration) and UV method for saliva (high concentration) seemed to be superior than the GC-MS measurement, further large-scale study with balance between sex and age groups should be performed to concrete our proof-of-concept study.

2) While the UV and colorimetric assessments of the enzymatic assay exhibit linearity, it is unusual that colorimetric data were only provided for Fig. 1A and 1B, and UV data for Fig. 1C and 1D. For completeness, also provide UV data (to accompany assessed samples for 1A and B) and colorimetry data (to accompany assessed samples for 1C and 1D).

<Response>

In order to show the correlation between enzymatic method and ultraviolet rays method for [C3-C6] assay in the saliva and linearity of C4 measurements obtained by these methods, we added NEW Fig 1 with three graphs showing the linearity for both enzymatic and UV data. After showing them, the former Fig 1 was changed to new Fig 2 for showing correlation between the enzymatic method and GC-MS method for the measurement of C2 and [C3-C6] in the GCF and saliva. Please see the Figs 1 and 2 with appropriate descriptions in the Results section.

To Reviewer #2

Really interesting work.

<Response>

Thank you for having your interests in our study, and we appreciate your valuable comments. We carefully reconsider our manuscript and sincerely respond to your comments.

But :

- Please do not use supporting information to store information that can bring important data to understand the problems and discovery, especially in results part, concerning S2B fig and the correlation with UV and color method (line 287)

<Response>

We reconstructed figures in order to emphasize the correlation between enzymatic method and ultraviolet rays method for [C3-C6] assay in the saliva and linearity of C4 measurements obtained by these methods. Thus, we added NEW Fig 1 with three graphs showing the linearity for both enzymatic and UV data, resulting the change of the former Fig 1 to be new Fig 2. Please see the Figs 1 and 2 with appropriate descriptions in the Results section.

- describe GC-MS method, by giving the conditions and material used for these analyses, instead of giving a reference

<Response>

We previously reported this method (see reference 15), but now added the precise description as following.

Internal standards acetate-d3, propionate-d3 and n-butyrate-d7 solutions were prepared in a final concentration of 5 mmol/L. Lastly, iso-butyrate-1-13C, n-valerate-1-13C, and caproate-2,2-d2 solutions were prepared in a final concentration of 10 mmol/L.

Saliva and GCF samples (450 µL) were transferred to 1.5 mL micro-tubes. Ten µL of each internal standard solution was added, and the mixture was vortexed for 10 s. The newly mixed samples were deproteinized with 10 µL of a 5- sulfosalicylic acid solution [1 g/mL of ultra-pure water (w/v)] and centrifuged at 2,000× g for 10 min at 4 ˚C. The supernatant was collected and transferred to a new test tube. Ten µL of hydrochloric acid (35 %) and 3 mL of diethyl ether were added, by vortexing the mixture (30 min) after each addition. Then, the final mixture was centrifuged at 1,200 × g for 10 min at room temperature. The ether phase was retrieved and transferred to a new test tube. Next, tert-butyldimethylchlorosilane (8 µL) was added and the new solution heated at 60 ˚C for 20 min. After heating, the solution was cooled down and transferred to a glass vial for GC-MS analysis.

A gas chromatography-mass spectrometry apparatus (GCMS-QP2010 Ultra; Shimadzu, Kyoto, Japan), equipped with an autosampler (AOC-5000; Shimadzu, Kyoto, Japan), was used. The GC-MS equipment was run in chemical ionization mode, with ammonia as the reagent gas. Chromatographic separation was carried out using a DB-5 MS column (30 m×0.25 mm I.D.×0.25 µm; Agilent Technologies, Santa Clara, CA, USA). Helium (1.5 mL/min) was used as the carrier gas.

These measurements were performed at the Kyoto Institute of Nutrition & Pathology (Kyoto, Japan). More detailed operating procedures have been reported previously [15].

- include lines 277-282 to the figure 1 legend. Same thing for line 299-310

<Response>

Thank you for your idea to improve the figures to be easily understood. According to your suggestion, we added these to the figures.

- line 329-333 : this conclusion seems to be unsuitable, since it depends the precision we are looking for. in fact, if in saliva the comparison between GC-MS and enzymatic methods lead to an almost good correlation, it is not the case in GCF.

<Response>

Since GCF was detected in low amounts and below the detection limit of the GC-MS method, we cannot say which method is better than the color method for GCF or the UV method for saliva in Fig. 2. However, these enzymatic measurements showed potential for clinical application in the future. The Discussion section was modified to reflect this point.

In general, the GC−MS method used to measure SCFAs is complicated and time-consuming [24], making it difficult to use in clinical practice. In contrast, the newly developed enzyme method can efficiently measure numbers of samples in a short procedure as shown in Table 2. It is also environmentally friendly because it does not use highly toxic organic solvents. The ability to distinguish between C2 and [C3-C6] simply by using the enzyme (Table 3) and the good correlation with the conventional standard GC-MS method using clinical samples such as GCF and saliva (Fig 2) indicates that this is a simple, inexpensive, and quick method that can be applied in clinical settings (Fig 3).

- in the conclusion, it should be interesting to give more information in term of saving money and time

<Response>

Thank you for suggesting the consideration for the issues for clinical application. Unfortunately, we did not compare the cost and time of them, but it is obvious that the GC-MS method is much more time-consuming and needs hazardous chemicals unfriendly for environment. We added short description to the Discussion section according to your comment as described above.

---

## [Editor Report · Decision Letter 1]

5 May 2022

Enzymatic measurement of short-chain fatty acids and application in periodontal disease diagnosis

PONE-D-21-34023R1

Dear Dr. Takashiba,

We’re pleased to inform you that your manuscript has been judged scientifically suitable for publication and will be formally accepted for publication once it meets all outstanding technical requirements.

Kind regards,

Joseph Banoub, Ph,D., D. Sc., FCIC, FRCS

Academic Editor

PLOS ONE
---

## [Editor Report · Acceptance letter]

7 Jul 2022

PONE-D-21-34023R1 

Enzymatic measurement of short-chain fatty acids and application in periodontal disease diagnosis 

Dear Dr. Takashiba:

I'm pleased to inform you that your manuscript has been deemed suitable for publication in PLOS ONE. Congratulations! Your manuscript is now with our production department. 

Kind regards, 

on behalf of

Dr. Joseph Banoub 

Academic Editor

PLOS ONE